# Failure Rate, Marginal Bone Loss, and Pink Esthetic with Socket-Shield Technique for Immediate Dental Implant Placement in the Esthetic Zone. A Systematic Review and Meta-Analysis

**DOI:** 10.3390/biology10060549

**Published:** 2021-06-18

**Authors:** Pilar Velasco Bohórquez, Roberta Rucco, Álvaro Zubizarreta-Macho, José María Montiel-Company, Susana de la Vega Buró, Esther Cáceres Madroño, Lara San Hipólito Marín, Sofía Hernández Montero

**Affiliations:** 1Department of Implant Surgery, Faculty of Health Sciences, Alfonso X el Sabio University, 28691 Madrid, Spain; mvelaboh@uax.es (P.V.B.); rrucc@myuax.com (R.R.); sdelabur@uax.es (S.d.l.V.B.); ecacemad@uax.es (E.C.M.); lsanhmar@uax.es (L.S.H.M.); shernmon@uax.es (S.H.M.); 2Department of Surgery, Faculty of Medicine and Dentistry, University of Salamanca, 37008 Salamanca, Spain; 3Department of Stomatology, Faculty of Medicine and Dentistry, University of Valencia, 46010 Valencia, Spain; jose.maria.montiel@uv.es

**Keywords:** socket shield, immediate implant, pink esthetic, implant failure, marginal bone loss

## Abstract

**Simple Summary:**

The socket-shield technique has been proposed for preserving the bone ridge and surrounding soft tissues with immediate implantation in the extraction socket, maintaining the buccal wall fragment of the dental root. However, the socket-shield technique has not been compared with the conventional technique for immediate dental implant placement in the esthetic zone regarding the failure rate, marginal bone loss, and pink esthetic. Therefore, it is necessary to develop a systematic review and meta-analysis that provides evidence associated with the prognosis when using the socket-shield technique compared to the conventional technique.

**Abstract:**

**Aim**: To compare the failure rate, marginal bone loss, and pink esthetic for the socket-shield technique and the conventional technique for immediate dental implant placement in the esthetic zone. **Material and methods**: A systematic literature review and meta-analysis, based on the Preferred Reporting Items for Systematic Reviews and Meta-Analyses (PRISMA) recommendations, of clinical studies that evaluated the failure rate, marginal bone loss, and pink esthetic with the socket-shield technique for immediate dental implant placement in the esthetic zone was performed. A total of 4 databases were consulted in the literature search: PubMed-MEDLINE, Scopus, Embase, and Web of Science. After eliminating duplicated articles and applying the inclusion criteria, 16 articles were selected for the qualitative and quantitative analysis. **Results**: Four randomized controlled trials, five prospective clinical studies, four retrospective studies, and three case series were included in the meta-analysis. The dental implant failure rate for the socket-shield technique for immediate dental implant placement was 1.37% (95% CI, 0.21–2.54%); however, no statistically significant differences between the conventional and socket-shield technique were found. The estimated mean difference in the marginal bone loss for the socket-shield technique was −0.5 mm (95% CI, −0.82 to −0.18) and statistically significant (*p* < 0.01), with a high heterogeneity (I^2^ = 99%). The mean pink esthetic score was 12.27 (Q test = 4.47; *p*-value = 0.61; I^2^ = 0%). The difference in pink esthetic between the conventional (*n* = 55) and socket-shield techniques (*n* = 55) for immediate dental implant placement was 1.15 (95% CI, 0.73–1.58; Q test = 8.88; *p* value = 0.11; I^2^ = 44%). The follow-up time was found to be significant (beta coefficient = 0.023; R^2^ = 85.6%; QM = 3.82; *p* = 0.049) for the PES for the socket-shield technique. **Conclusions**: Within the limitations of this systematic review with meta-analysis, the dental implant failure rate did not differ between the socket-shield technique and conventional technique for immediate implant placement in the esthetic zone. However, a lower marginal bone loss and higher pink esthetic scores were found for the socket-shield technique compared to the conventional technique.

## 1. Introduction

Dental extractions can cause volumetric changes in bone tissues characterized by the resorption of alveolar bone, especially the buccal bone wall, with a consequent retraction of the related soft tissues. Additionally, multiple-tooth extractions can lead to the loss of the dental papilla [1,2]. Biological mechanisms involved in the healing of the periodontal tissues after tooth extraction can cause the loss of the periodontal ligament and its vascular support [3,4]. The aforementioned physiological process can lead to esthetic problems that are difficult to resolve by methods of restoration that are able to preserve the emergence profile, especially in the anterior region. Therefore, preserving and maintaining the bone anatomy and soft tissue architecture in the anterior region is essential for maintaining esthetics in implant-supported restorations [5]. Different techniques and materials have been proposed to prevent the bone resorption; however, immediate dental implant placement and alveolar preservation procedures have been recommended [6]. Osseointegration has been defined as a direct and functional connection between bone and an artificial implant. Both the macroscopic and microscopic characteristics of dental implants could influence the success of these procedures [7]. Unfortunately, the abovementioned regenerative approaches cannot prevent the esthetic implications of physiological bone resorption or the physiological consequences of the reduced vascular supply after tooth extraction [2,8]. Therefore, the socket-shield technique was proposed for preserving the bone ridge and surrounding soft tissues, with immediate implantation in the extraction socket, maintaining the buccal wall fragment of the dental root [3,5]. The periodontal ligament and associated blood vessels avoid the initiation of buccal osteoclastic activity, and bone resorption and the contraction of surrounding soft tissues are prevented [5,9]. The socket-shield technique has shown a success rate of 96.5% [2]; however, teeth affected by periodontal disease, vertical or horizontal root fractures under the bone ridge, and internal root resorption can influence the prognosis after the placement of the dental implant and require further research [5]. In addition, the following clinical complications have been associated with the socket-shield technique: a lack of osteointegration of the dental implant, infections, and the mobilization, migration, and resorption of the root fragment [6]. An acceptable pink esthetic around the dental implant is generally demanded by the patient. The pink esthetic score (PES) is an index used to evaluate the soft tissue characteristics around single-tooth implant crowns. It takes into account the soft tissue level, soft tissue contour, alveolar process deficiency, soft tissue color, and texture. The maximum achievable PES is 14, with a 0–1–2 scoring system; 0 is the lowest and 2 is the highest value [10].

The aim of this systematic review with meta-analysis was to analyze the failure rate, marginal bone loss, and pink esthetic with the socket-shield technique compared to those with the conventional technique for immediate dental implant placement in the esthetic zone. The null hypothesis (H_0_) was that there would be no difference in the dental implant failure rate, marginal bone loss, and pink esthetic between the two techniques.

## 2. Materials and Methods

### 2.1. Study Design

A bibliographic search was conducted following the PRISMA (Preferred Reporting Items for Systemic Reviews and Meta-Analyses http://www.prisma-statement.org (accessed on 15 April 2020)) guidelines for systematic reviews and meta-analyses (INSPLAY registration number: INPLASY202110058). The review also fulfilled the PRISMA 2009 Checklist [11].

### 2.2. Question of Interest

The PICO (population, intervention, comparison, and outcome) question was ‘Whatis the dental implant failure rate, marginal bone loss, and pink esthetic of socket-shield technique for dental immediate implant placement compared to conventional dental immediate placement in the esthetic zone?’ with the following components: population—patients treated with the socket-shield technique for immediate dental implant placement; intervention—socket-shield technique for immediate dental implant placement in the esthetic zone; comparison—conventional immediate dental implant placement in the esthetic zone; and outcome—the dental implant failure rate, marginal bone loss, and pink esthetic.

### 2.3. Databases and Search Strategy

An electronic search was conducted in the following databases: PubMed, Scopus, Embase, Web of Sciences, and OpenGrey (www.opengrey.eu, accessed on 15 April 2020). The search covered all the literature published internationally up to July 2020. The search included fifteen medical subject heading (MeSH) terms: ‘socket shield technique’; ‘root membrane’; ‘ridge preservation’; ‘tooth socket’; ‘tooth extraction’; ‘tooth root’; ‘partial extraction therapy’; ‘anterior implant’; ‘immediate implant’; ‘immediate dental implant loading’; ‘dental implants’; ‘single-tooth’; ‘dental implantation’; ‘endosseous’; and ‘aesthetic area implant’. The Boolean operators applied were ‘OR’ and ‘AND’. The search terms were structured as follows: (“socket-shield technique”) OR (“root-membrane”) OR (“ridge preservation”) OR (“tooth Socket”)OR (“tooth Extraction”) OR (“tooth Root”) OR (“partial extraction therapy”) AND (“anterior implant”) AND (“immediate implant”) OR (“immediate Dental Implant Loading”) OR (“dental Implants, Single-Tooth”) OR (“dental Implantation”), (“endosseous”) OR (“esthetic area implant”). Two researchers (R.R. and Á.Z.-M.) conducted the database searches in duplicate independently.

### 2.4. Study Selection

Titles and abstracts were selected with two authors applying inclusion and exclusion criteria (Á.Z.-M. and J.M.M.-C.).

One researcher (R.R.) extracted data for the relevant variables. The systematic review was carried out (S.T.G.) and subsequent meta-analysis was performed by two researchers not involved in the selection process (Á.Z.-M. and J.M.M.-C.).

Inclusion criteria: studies recorded in databases as prospective randomized clinical trials (RCTs), retrospective studies, and case series from three patients. The review was not restricted to only RCTs due to the paucity of studies with such an experimental design and with external validity, but also to provide a complete picture of the topic. Studies that analyzed clinical and/or radiographic marginal bone loss, implant failure rates, soft tissue results, and pink esthetic scores after immediate dental implant placement in the esthetic zone using the socket-shield technique were included. Studies with samples of patients aged 18 years old or over, patients treated with the socket-shield technique for immediate dental implant placement in the esthetic zone, and follow-up periods of at least 3 months were included. No restriction was placed on the year of publication or language.

Exclusion criteria: systematic literature reviews, clinical cases, case series up to three patients and editorials; studies including patients under the age of 18 years old; studies with samples of three or fewer patients.

### 2.5. Data Extraction and Study Outcomes

The following data were extracted from each article: the author and year of publication, study type, sample size, follow-up in months, marginal bone loss, implant failures, soft tissue results, and pink esthetic scores. Data were extracted in duplicate (R.R. and S.H.M.) using predefined Excel spreadsheets.

### 2.6. Methodological Quality Assessment

The risk of bias in the studies selected for review was assessed two authors (Á.Z.-M. and J.M.M.-C.) using the Jadad scale for assessing the methodological quality of clinical trials. The Jadad scale consists of five items that evaluate the randomization, researcher and patient blinding, and description of losses during follow-up, producing scores of 0–5; scores less than 3 are considered indicative of low quality [12]. The level of agreement between evaluators was determined using Kappa scores.

### 2.7. Quantitative Synthesis—Meta-Analysis

The statistical data collection and analysis were conducted by two authors (Á.Z.-M. and J.M.M.-C.). The studies included for the meta-analysis were combined using a random-effects model with various methods according to the estimated effect size. The inverse-variance method was used to estimate the root apex location success rate, the Mantel–Haenszel method for the odds ratio (OR), and the inverse-variance method for the mean difference. For all the estimated variables, the 95% confidence intervals were calculated. The heterogeneity between the combined studies was assessed using the Q test (*p*-value < 0.05) and quantified with the I^2^, considering slight heterogeneity if it was 25–50%, moderate if 50–75%, and high if > 75%. Statistical significance was tested for using the Z test (*p*-value < 0.05). The meta-analyses are represented with forest plots. The publication bias was assessed using the trim and fill adjustment method, and is represented with Funnel plots. The R software was employed for the meta-evidence analysis.

## 3. Results

### 3.1. Flow Diagram

The initial electronic search was performed in June of 2020: 21 articles in PubMed, 31 in Web of Sciences, 17 in Embase, 10 in Scopus, and two in gray literature were identified. Of the total of 81 studies, 14 were discarded due to being duplicates. After screening the titles and abstracts, a further 26 were eliminated, leaving a total of 41. A further 13 were rejected, as they failed to fulfil the following inclusion criteria: including survival rate data, including pink esthetic data, and presenting a minimum follow-up time of 3 months. A final total of 16 articles were included in the qualitative and quantitative synthesis, as these included all the data and variables required (Figure 1).

### 3.2. Qualitative Analysis

Of the 16 articles included, 4 were randomized clinical trials [4,6,13,14], 5 were prospective clinical trials [15,16,17,18,19], 4 were retrospective studies [2,7,20,21], and 3 were case series [8,22,23]. In addition, 6 articles compared the outcomes of conventional immediate dental implant placement versus such placement using the socket-shield technique [4,6,12,13,14,15]. The sample sizes of the studies selected in the present meta-analysis range from 4 in the study by Nguyen et al., 2019 [8] to the high figure of 250 in Siormpas’ study, 2018 [2], with the subject ages ranging from 18 [2] to 87 [8] years, and the follow-up times from 3 [21] to 120 [2] months. The results are presented in Table 1.

### 3.3. Quality Assessment

The results of the methodological quality assessment using the Jadad scale are shown in Table 2. The Jadad scale was “Not applicable” to seven articles because they were retrospective studies [2,7,19,20] and a case series [8,21,22], and the authors of these articles were not blinded, nor were the studies randomized. Some randomization and blind procedures were “Not available” and, hence, the corresponding studies were not assigned scores. Two articles [4,12] received scores of 4, indicating high methodological quality. Again, the quality was most frequently compromised by failure to fulfil criteria related to the subject, treatment, language, or measurement blinding.

### 3.4. Quantitative Analysis

#### 3.4.1. Failure Rate

Sixteen studies including a total of 599 implants, with different follow-up periods ranging from 3 [21] to 120 [2] months, were combined using a random-effects model with the inverse-variance method. The rate of failure for the socket-shield technique for immediate dental implant placement in the esthetic zone was estimated to be 1.37%, with a 95% confidence interval of 0.21–2.54%. The meta-analysis showed no heterogeneity between the combined studies (Q-test = 4.98; *p*-value = 0.992; I^2^ = 0%). (Q test = 32.4; *p*-value = 0.070; I^2^ = 32.1%) (Figure 2).

The study follow-up time was found not to be a significant variable (beta coefficient = 0.0005) in a meta-regression with the mixed-effects model (test of moderators (R^2^ = 0%; QM = 2.23; *p* = 0.134)) to estimate the percentage of dental implant failure with the socket-shield technique for immediate dental implant placement in the esthetic zone. The follow-up time does not seem to affect the prognosis after the immediate dental implant placement regardless of the placement technique.

Six studies [4,6,12,13,14,15] compared the rates of dental implant failure for the socket-shield technique (*n* = 75) and the conventional technique (*n* = 81); however, no statistically significant differences (odds ratio = 1.09; *p* value = 1; I^2^ = 0%) were observed (Figure 3).

#### 3.4.2. Publication Bias

Seven studies were added to the 16 studies initially combined, using the trim and fill method to obtain symmetry in the funnel plot. The dental implant failure rate for the socket-shield technique, adjusted using the inverse-variance random-effects model, was 1.78% (95% CI, 0.73–2.83), showing no difference with respect to the initial 1.37 (Figure 4).

#### 3.4.3. Marginal Bone Loss

Three studies compared the marginal bone loss for the conventional immediate dental implant placement (*n* = 58) and the socket-shield technique (*n* = 58). The estimated mean difference was −0.5 mm (95% CI, −0.82 to −0.18) and statistically significant (*p* < 0.01), and the heterogeneity was high according to the meta-analysis (I^2^ = 99%) (Figure 5.

The study follow-up time was not found to be a significant variable (beta coefficient = 0.0094) in a meta-regression with the mixed-effects model (R^2^ = 0%; test of moderators QM = 0.090; *p* = 0.764) to estimate the mean difference in bone loss between the two techniques.

#### 3.4.4. Publication Bias

Two studies were added to the three studies initially combined, using the trim and fill method to obtain symmetry in the funnel plot. The estimated mean difference in marginal bone loss, adjusted using the inverse-variance random-effects model, was −0.15 mm (95% CI, −0.43 to 0.13), showing a significant difference with respect to the initial −0.50 mm (Figure 6).

#### 3.4.5. Pink Esthetic Score

Scores from three studies with pink esthetic score (PES) measurements taken at different points during the follow-up have been included. The mean PES obtained by combining the studies with the random-effects model (inverse variance) was 12.27 (range, 12.12–12.41). Heterogeneity was not detected (Q test = 4.47; *p* value = 0.61; I^2^ = 0%) (Figure 7).

Two studies with three measurements throughout the follow-up period (ranging from 3 to 40 months) compared the PESs for the conventional immediate dental implant placement (*n* = 55) and the socket-shield technique (*n* = 55). The meta-analysis (random-effects model combined with the inverse-variance method) estimated a mean difference between the techniques of 1.15 (95% CI, 0.73–1.58). There was slight heterogeneity between the studies (Q test = 8.88; *p*-value = 0.11; I^2^ = 44%). The PES for the socket-shield technique showed a difference of 1.15 points with respect to that for the conventional technique (Figure 8).

The follow-up time was found to be a significant variable (beta coefficient = 0.023) in a meta-regression with the mixed-effects model (R^2^ = 85.6%; QM = 3.82; *p* = 0.049) for estimating the mean difference in PES between the conventional immediate dental implant placement and placement using the socket-shield technique. The difference in PES increased by 0.02 points/month for the socket-shield technique with respect to the conventional method (Figure 9).

#### 3.4.6. Publication Bias

Three studies were added to the seven studies initially combined, using the trim and fill method to obtain symmetry in the funnel plot. The estimate of the mean PES adjusted by the inverse random-effects model of the variance was 12.30 (95% CI, 12.16–12.44), not showing a difference with respect to the initial 12.27 (Figure 10).

## 4. Discussion

This systematic review with meta-analysis was centered on the dental implant failure rate, marginal bone loss, and pink esthetic for a single immediate dental implant in the esthetic zone through the conventional or socket-shield technique. The results obtained in the present study refute the null hypothesis (H_0_) stating that there was no statistically significant difference in the marginal bone loss and pink esthetic between the socket-shield and conventional techniques for immediate dental implant placement in the esthetic zone. However, there was no statistically significant difference in the dental implant failure rate between the techniques.

A mean dental implant failure rate of 1.37% (95% CI: 0.21–2.54%) was found, in this systematic review with meta-analysis, for the socket-shield technique; moreover, the studies with the highest sample sizes and longest follow-ups [2,20] showed high rates of dental implant survival. Gluckman et al. analyzed 128 immediate dental implants performed in the esthetic zone with the socket-shield technique with at least four years of follow-up and reported a dental implant survival rate of 96.1% [20], and Siormpas et al. included 250 immediate dental implants performed with the socket-shield technique with a follow-up of 10 years and reported a dental implant survival rate of 98%; however, some eventual complications such as infections of the root membrane, internal and external exposures, and migration of the dental implant were reported [2].

Hürzeler et al. established that the main objective of the socket-shield technique for immediate dental implant placement is preserving the buccal bone plate, which could influence the esthetic results [21]; therefore, most of the dental implants in the selected articles were placed in the anterior maxillary. However, this location usually presents a narrow buccal cortical plate, which increases the risk of marginal bone resorption after tooth extraction [22]. Moreover, Tsigarida et al. reported that most of the buccal bone walls in anterior maxillary teeth are narrower than 1 mm at the coronal third, and buccal bone walls wider than 2 mm were only measured in the middle third of canine and premolar teeth and in the apical third of every tooth [23]. However, the thickness of the marginal bone crest around teeth can remain stable due to the vascular supply from the periodontal vessels [24], although a thin marginal bone crest around dental implants can be reabsorbed, leading to the exposure of the rough surface of the dental implant [24] because the fasciculated bone of the internal portion of the alveolus is usually reabsorbed after tooth extraction without the adjacent periodontal tissues [24], leading to an unaesthetic effect due to the retraction of the buccal mucosa after buccal bone resorption [4]. Therefore, the maintenance of the periodontal tissues is still a concern, which could be solved by the preservation of the buccal fragment of the dental root [6]. In addition, Mitsias et al. reported an absence of buccal bone loss at 5 years follow-up using the socket-shield technique for dental immediate implant placement in the esthetic zone [25], and Baümer et al. highlighted the biological effect of the socket-shield technique [26]. In addition, some authors have reported the influence of grafting materials for filling the space between the root fragment and the dental implant. Gluckman et al. reported that the space between the implant and the buccal portion of the root fragment should always be filled with graft material [20]. Habashneh et al. and Bramanti et al. recommend filling the space with a heterologous graft material to improve the healing process and to further reduce bone resorption and facilitate bone–implant contact [4,22]. However, Siormpas et al. suggested that it is not necessary to graft the space between the residual buccal root fragment and the dental implant [7]. This concept is supported by recent histological data showing that, without the use of biomaterials, new bone grows in the space between the dentin fragment and the dental implant [4].

Blaschke and Schwass published a systematic review without meta-analysis and highlighted promising outcomes for the socket-shield technique; however, they also mentioned the limited data available related to well-designed prospective randomized controlled studies, which tend not to report the long-term outcomes for the socket-shield technique [27]. Ogawa et al. also published a systematic review without meta-analysis and reported 90.5% implant survival and a low failure rate [28]; however, the present study only showed 1.37% implant failure.

The maintenance of the marginal bone crest associated with the socket-shield technique for dental immediate implant placement could influence the high PES observed. This esthetic visual index was selected out of eight esthetic evaluation indices, as it was deemed to be the most reliable and valid; PES is therefore often used to analyze the pink esthetic around dental immediate implants [29].

The PES measurement index takes into account the mesial and distal papilla insertion level, the soft tissue level and contour, the alveolar process deficiency and the soft tissue color and texture. The high mean PESs (12.27 (range, 12.12–12.41)) shown in this systematic review with meta-analysis could be attributable to the few volumetric alterations of the soft tissues and, hence, to the maintained marginal bone crest surrounding the immediate dental implants with the socket-shield technique. Moreover, Baümer et al. reported minimal changes associated with the gingival contour, few recessions observed at both the immediate dental implants and the neighboring teeth, and little marginal bone loss, showing compatibility with peri-implant health [19]. Additionally, Hinze et al. observed volumetric changes minor to 0.5 mm in all cases with a follow-up of 3 months [21].

From the literature reviewed, the socket-shield technique for immediate dental implant placement in the esthetic zone seems to be a successful and minimally invasive technique, although more long-term and better-designed studies are needed. As for the limitations of this systematic review with meta-analysis, there was a risk of articles related to the selection criteria not being found, although this risk was reduced by searching in 4 databases. In addition, most of studies presented poor methodological quality with scores lower than 3 on the Jadad scale.

## 5. Conclusions

Within the limitations of this systematic review with meta-analysis, the rate of dental implant failure did not differ between the socket-shield technique and conventional technique for immediate implant placement in the esthetic zone. However, lower marginal bone loss and higher pink esthetic scores were found for the socket-shield technique.

## Figures and Tables

**Figure 1 biology-10-00549-f001:**
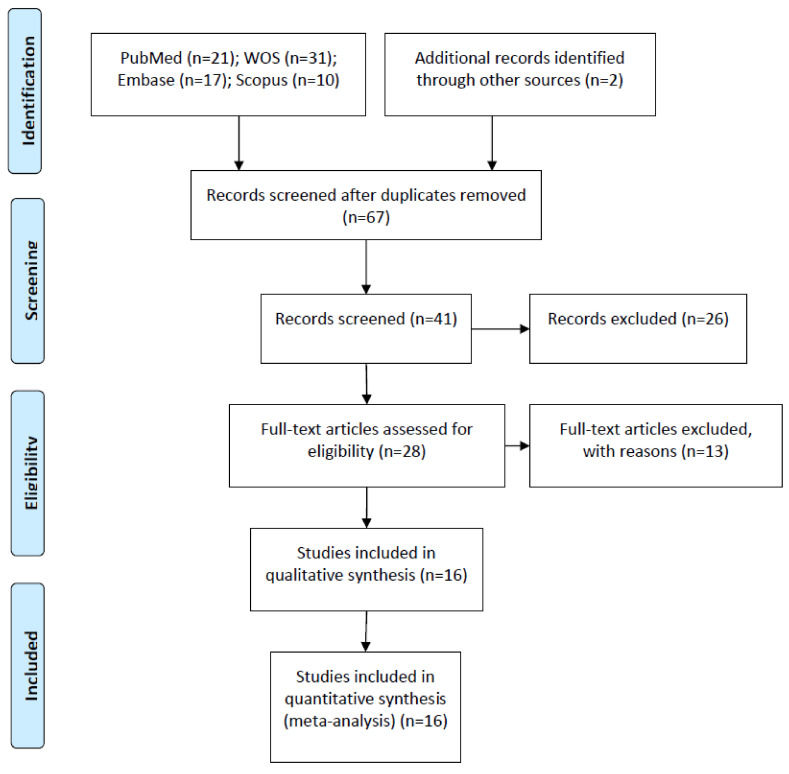
Preferred Reporting Items for Systematic Reviews and Meta-Analyses (PRISMA) flow diagram.

**Figure 2 biology-10-00549-f002:**
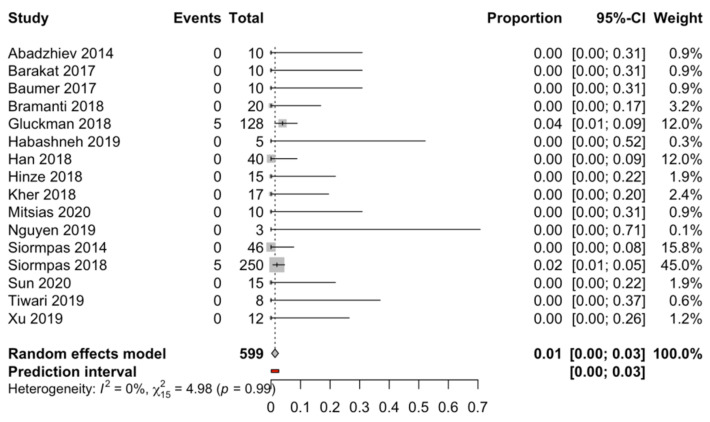
Forest plot of the meta-analysis of implant failure with immediate dental implant placement using the socket-shield technique.

**Figure 3 biology-10-00549-f003:**
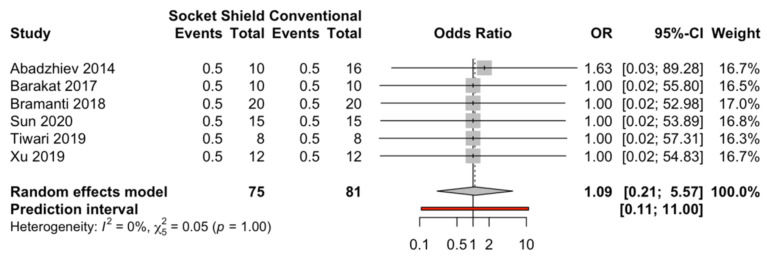
Forest plot of the rates of dental implant failure (odds ratio) for immediate dental implant placement using the socket-shield technique in the esthetic zone and conventional immediate dental implant placement.

**Figure 4 biology-10-00549-f004:**
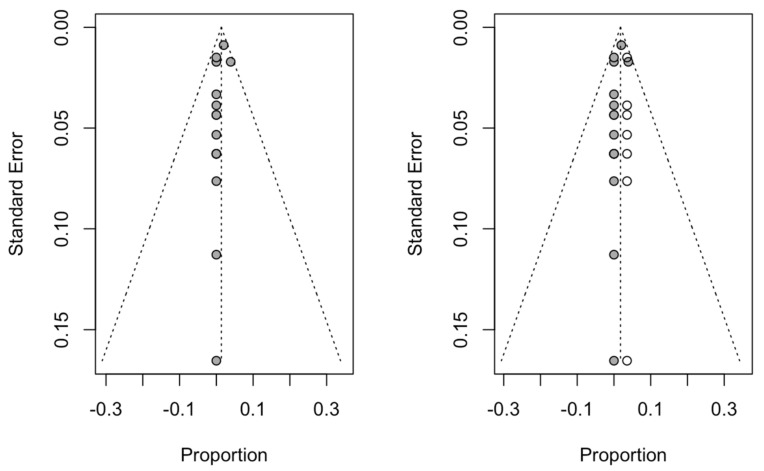
Initial funnel plot and plot after trim and fill adjustment of the dental implant failure of the socket-shield technique for dental immediate implant placement in the esthetic zone.

**Figure 5 biology-10-00549-f005:**
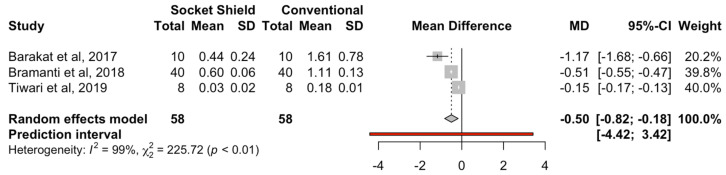
Forest plot of the mean difference in marginal bone loss (mm) between immediate dental implant placement using the socket-shield technique and conventional placement technique.

**Figure 6 biology-10-00549-f006:**
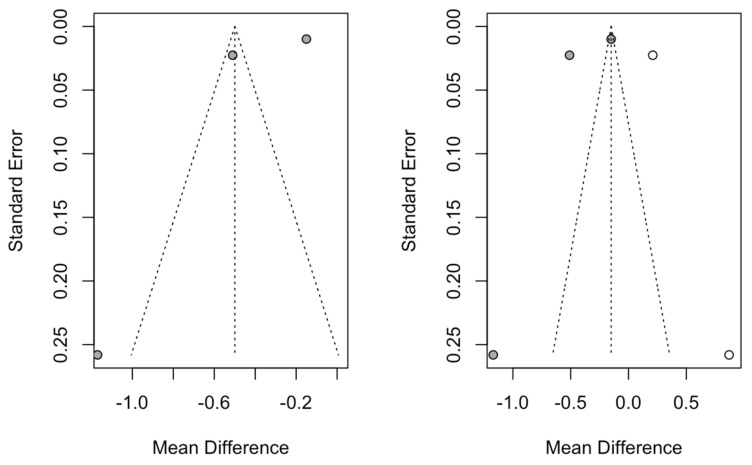
Initial funnel plot and plot after trim and fill adjustment of the mean difference in bone loss (mm) for the immediate dental implant placement in the esthetic zone using the socket-shield technique.

**Figure 7 biology-10-00549-f007:**
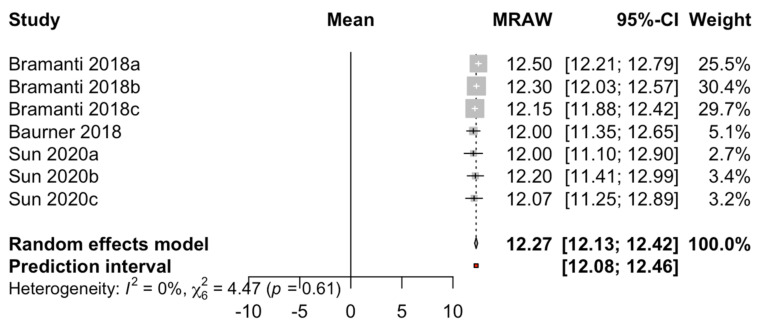
Forest plot of the mean PES of immediate dental implant placement using the socket-shield technique in the esthetic zone.

**Figure 8 biology-10-00549-f008:**
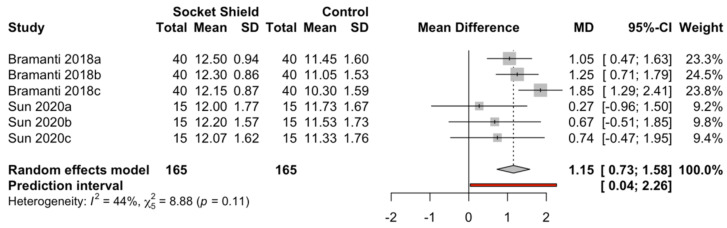
Forest plot of the mean difference in PES between immediate dental implant placement using the socket-shield technique and conventional technique in the esthetic zone.

**Figure 9 biology-10-00549-f009:**
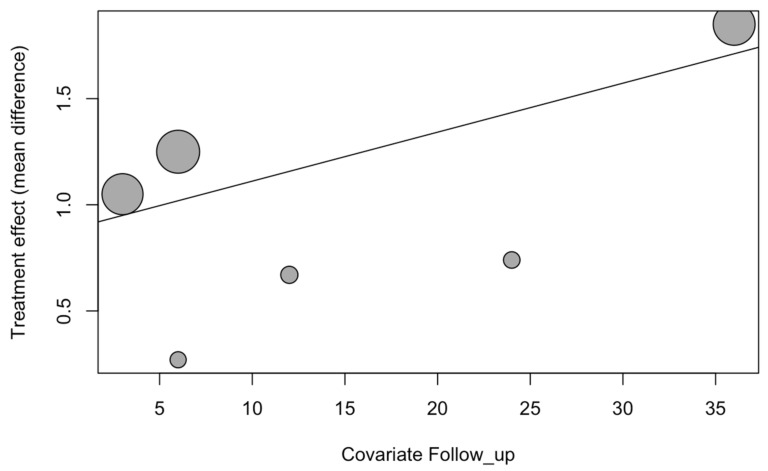
Bubble plot of follow-up time as a covariate of the PES for the immediate dental implant placement using the socket-shield technique in the esthetic zone.

**Figure 10 biology-10-00549-f010:**
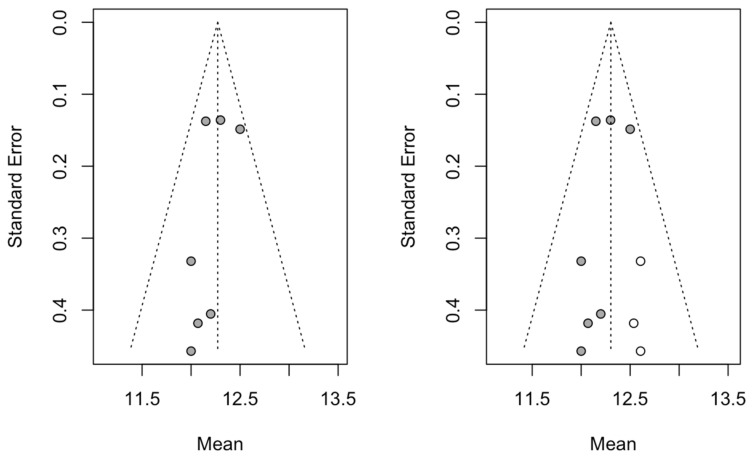
Initial funnel plot and plot after trim and fill adjustment of the mean difference in PES for the immediate dental implant placement using the socket-shield technique in the esthetic zone.

**Table 1 biology-10-00549-t001:** Assessment of methodological quality according to the Jadad scale.

Jadad Criteria						
Author/Year	Is the Study Described as Randomized?	Is the Study Described as Double-Blinded?	Was There a Description of Withdrawals and Dropouts?	Was the Method of Randomization Adequate?	Was the Method of Blinding Appropriate?	Score
Abadzhiev et al., 2014 [15]	0	0	0	0	0	0
Barakat et al., 2017 [6]	1	0	0	NAv	0	1
Baumer et al., 2017 [19]	NA	NA	NA	NA	NA	NA
Bramanti et al., 2018 [4]	1	1	0	1	1	4
Gluckman et al., 2018 [20]	NA	NA	NA	NA	NA	NA
Habasneh et al., 2019 [22]	NA	NA	NA	NA	NA	NA
Han et al., 2018 [5]	0	0	0	0	0	0
Hinze et al., 2018 [21]	NA	NA	NA	NA	NA	NA
Kher et al., 2018 [18]	0	0	0	0	0	0
Mitsias et al., 2020 [16]	0	0	0	0	0	0
Nguyen et al., 2019 [8]	NA	NA	NA	NA	NA	NA
Siormpas et al., 2014 [7]	NA	NA	NA	NA	NA	NA
Siormpas et al., 2018 [2]	NA	NA	NA	NA	NA	NA
Sun et al., 2020 [12]	1	1	0	1	1	4
Tiwari et al., 2019 [13]	1	0	0	Nav	0	1
Xu et al., 2019 [14]	NAv	NAv	NAv	NAv	NAv	NAv

NA: Not applicable; NAv: Not available.

**Table 2 biology-10-00549-t002:** Qualitative analysis of articles included in the systematic review.

Author/Year	Study Type	Sample (*n*)	Follow-Up Time (Months)	Dental Implant Failure Rate	Marginal Bone Loss	Soft Tissue Results and Pink Esthetic
Abadzhiev et al., 2014 [15]	Prospective clinical trial	26 dental implants (25 patients 20–64 years old)	24	0/16 CIIP 0/10 SST	CIIP: 12% bone loss (5 mm) SST: 2% bone loss (0.8 mm)	CIIP: 12% attached gingiva loss (5 mm) SST: 2% attached gingiva loss (0.8 mm)
Barakat et al., 2017 [6]	RCT	20 dental implants (20 patients 20–50 years old)	7	0/10 CIIP 0/10 SST	CIIP: 1.61 ± 0.78 mm vertical bone loss SST: 0.44 ± 0.24 mm vertical bone loss	CIIP: 2.12 ± 0.64 mm probing depth SST: 1.73 ± 0.28 mm probing depth
Baumer et al., 2017 [19]	Retrospective study	10 dental implants (10 patients)	51–63	0/10 SST	0.33 ± 0.43 mm mesial and 0.17 ± 0.36 mm distal marginal bone loss	SST: −0.37 ± 0.18 mm loss of buccal tissue and −0.33 ± 0.23 mm mid-facial recession Pink aesthetic score: 12
Bramanti et al., 2018 [4]	RCT	40 dental implants (40 patients)	36	0/20 CIIP 0/20 SST	CIIP: 1.11 ± 0.13 mm marginal bone loss SST: 0.60 ± 0.06 mm marginal bone loss	CIIP: Pink aesthetic score: 10.30 ± 2.53 SST: Pink aesthetic score: 12.15 ± 0.76
Gluckman et al., 2018 [20]	Retrospective study	128 dental implants (128 patients 24–71 years old)	48	5/128 SST	NAv	NAv
Habasneh et al., 2019 [22]	Case series	5 dental implants (5 patients 20–54 years old)	12	0/5 SST	NAv	NAv
Han et al., 2018 [5]	Prospective clinical trial	40 dental implants (30 patients 20–82 years old)	12	0/40 SST	NAv	0/40 SST
Hinze et al., 2018 [21]	Case series	17 dental implants (15 patients 26–66 years old)	3	0/17 SST	NAv	SST: 0.17 ± 0.67 mm change in the gingival margin SST: 8/15 patients suffer recession SST: 0.31 ± 0.64 mm mesial papilla height change and −0.38 ± 0.57 mm distal papilla height change
Kher et al., 2018 [18]	Prospective clinical trial	21 dental implants (17 patients 26–66 years old)	12–42	0/21 SST	NAv	SST: Pink aesthetic score: 12
Mitsias et al., 2020 [16]	Prospective clinical trial	10 dental implants (10 patients)	42	0/10 SST	NAv	SST: 0.19 mm (0.10–0.28 mm) mid-facial recession
Nguyen et al., 2019 [8]	Case series	4 dental implants (3 patients 62–87 years old)	24–72	0/4 SST	0.1 ± 0.2 mm marginal bone loss	SST: No changes in soft tissue dimensions
Siormpas et al., 2014 [7]	Retrospective study	46 dental implants (46 patients 28–70 years old)	24–60	0/46 SST	0.18 ± 0.09 mm mesial and 0.21 ± 0.09 mm distal marginal bone loss	NAv
Siormpas et al., 2018 [2]	Retrospective study	250 dental implants (182 patients 18–83 years old)	120	5/250 SST	NAv	NAv
Sun et al., 2020 [12]	RCT	30 dental implants (30 patients	24	0/15 CIIP 0/15 SST	NAv	CIIP: Pink aesthetic score: 11.33 ± 1.76 SST: Pink aesthetic score: 12.07 ± 1.62
Tiwari et al., 2019 [13]	RCT	16 dental implants (16 patients)	12	0/8 CIIP 0/8 SST	CIIP: 0.188 ± 0.013 mm marginal bone loss SST: 0.030 ± 0.025 mm marginal bone loss	CIIP: Labial bone thickness: 0.988 ± 0.173 mm SST: Labial bone thickness: 1.145 ± 0.277 mm
Xu et al., 2019 [14]	Prospective clinical trial	24 dental implants (24 patients)	12	0/12 CIIP 0/12 SST	NAv	SST higher PES than CIIP

CIIP: conventional immediate dental implant placement; SST: socket-shield technique; RCT: randomized controlled trial; NAv: not available.

## Data Availability

Data available on request due to restrictions, e.g., privacy and ethical.

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
