# Peer review of "Failure Rate, Marginal Bone Loss, and Pink Esthetic with Socket-Shield Technique for Immediate Dental Implant Placement in the Esthetic Zone. A Systematic Review and Meta-Analysis"

_biology, 2021, doi:10.3390/biology10060549_

Round 1
Reviewer 1 Report
The definitions of this work are appropriate. The analysis of the results shows that the differences are not statistically significant. At the same time, my doubts are raised by the fact that analyzed in one group are papers assessing the condition of tissues after 3 months and several years. Obviously, in the initial period, the condition of soft tissues is different than after a few years. It seems reasonable to increase the assessment threshold to, for example, 12 months. Then we are sure that the procedures performed during the installation of the implant and prosthetic restoration do not affect the assessment. These aspects are included in the discussion, although in my opinion insufficiently described. I would suggest that the authors pay more attention to this aspect. after e.g. 4 years, the shape of the prosthetic restoration is also important, which the authors do not mention.
Author Response
Dear Reviewer 1:
I’m pleased to resubmit the manuscript of the work entitled, “Failure Rate, Marginal Bone Loss and Pink Esthetic of Socket-Shield Technique for Immediate Dental Implant Placement in the Esthetic Zone. A Systematic Review and Meta-Analysis“
Reviewer 1: I don't feel qualified to judge about the English language and style
Response: In order to adapt to the reviewer's 11 comments, we have send the manuscript to the English Editing Service of MDPI. We attached the Certificate.
Reviewer 1: At the same time, my doubts are raised by the fact that analyzed in one group are papers assessing the condition of tissues after 3 months and several years. Obviously, in the initial period, the condition of soft tissues is different than after a few years. It seems reasonable to increase the assessment threshold to, for example, 12 months. Then we are sure that the procedures performed during the installation of the implant and prosthetic restoration do not affect the assessment. These aspects are included in the discussion, although in my opinion insufficiently described. I would suggest that the authors pay more attention to this aspect.
Response: In order to adapt to the reviewer's 1 comments, we clarify that the article: Hinze M, Janousch R, Goldhahn S, Schlee M. Volumetric alterations around single-tooth implants using the socket-shield technique: preliminary results of a prospective case series. Int J Esthet Dent. 2018, 13, 146-170, is the only one which assess the condition of tissues for 3 months, and we highlight that the article Moskow BS. The Response of the Gingival Sulcus to Instrumentation: A Histological Investigation. 2. Gingival curettage J Periodontol. 1964, 35, 112-126, reports that the stabilization of soft tissues occurs after 3 months.
Reviewer 1: after e.g. 4 years, the shape of the prosthetic restoration is also important, which the authors do not mention.
Response: In order to adapt to the reviewer's 1 comments, we clarify that unfortunately, the authors of the selected articles do not provide information related to the prosthetic restoration.
We take this opportunity to thank the recommendations and suggestions made by the reviewer to improve the document.
Yours sincerely,
Reviewer 2 Report
The manuscript submitted to Biology entitled “Failure Rate, Marginal Bone Loss and Pink Esthetic of Socket-Shield Technique for Immediate Dental Implant Placement in the Esthetic Zone. A Systematic Review and Meta-Analysis” is a systematic review with meta-analysis which aim to analyze the failure rate, the marginal bone loss and the pink esthetic between the socket shield technique and the conventional technique for immediate dental implant placement in the esthetic zone.
On my opinion the article is interesting, well written, with good English. Anyway, there are some minor issues to address.
- English language: Minor corrections needed.
- Abstract: Please structure the abstract to attract the reader's attention.
- Introduction: My main suggestion is to include a brief sentence on osseointegration and factors that can affect it: <<Osseointegration has been defined as a direct and functional connection between bone and an artificial implant. Both macroscopic and microscopic characteristics of dental implants could influence the success of these procedures [doi:10.23812/20-96-L-53]>>.
- Materials and Methods: This section has been properly prepared.
- Results: This section has been properly prepared.
- Discussion: Can the distance between the vestibular residual root and the implant fixture impact the success of the procedure? Please discuss briefly.
What are the results of other systematic reviews on the subject? - Figures: Please improve quality and resolution.
- Abbreviations: Insert a summary of abbreviations used in the text prior to “Reference” section.
After making the indicated changes, the article will be suitable for publication.
Thanks for the opportunity to review this manuscript.
Author Response
Dear Ms Fancy Yan:
I’m pleased to resubmit the manuscript of the work entitled, “Failure Rate, Marginal Bone Loss and Pink Esthetic of Socket-Shield Technique for Immediate Dental Implant Placement in the Esthetic Zone. A Systematic Review and Meta-Analysis“
Reviewer 2: English language and style are fine/minor spell check required
Response: In order to adapt to the reviewer's 2 comments, we have send the manuscript to the English Editing Service of MDPI. We attached the Certificate.
Reviewer 2: English language: Minor corrections needed.
Response: In order to adapt to the reviewer's 2 comments, we have send the manuscript to the English Editing Service of MDPI. We attached the Certificate.
Reviewer 2: Abstract: Please structure the abstract to attract the reader's attention.
Response: In order to adapt to the reviewer's 2 comments, we have structured the Abstract.
Reviewer 2: Introduction: My main suggestion is to include a brief sentence on osseointegration and factors that can affect it: <<Osseointegration has been defined as a direct and functional connection between bone and an artificial implant. Both macroscopic and microscopic characteristics of dental implants could influence the success of these procedures [doi:10.23812/20-96-L-53]>>.
Response: In order to adapt to the reviewer's 2 comments, we have added the sentence and reference.
Reviewer 2: Discussion: Can the distance between the vestibular residual root and the implant fixture impact the success of the procedure? Please discuss briefly.
Response: In order to adapt to the reviewer's 2 comments, we clarify that the literature describes that, in the socket-shield technique, the implant can be placed in direct contact with the root fragment or leaving a small space. In recent years, different research groups have reviewed the technique originally introduced by Hurzeler and have proposed variants. Gluckman et al. reported that, if present, the space between the implant and the buccal portion of the root should always be filled with graft material (Gluckman H, Salama M, Du Toit J. A retrospective evaluation of 128 socket-shield cases in the esthetic zone and posterior sites: Partial extraction therapy with up to 4 years follow-up. Clin Implant Dent Relat Res. 2018, 20, 122-129.). Habashneh et al. and Bramanti et al. filled the present space with heterologous graft material to improve the healing process and to further reduce ridge resorption and facilitate bone-implant contact (Bramanti E, Norcia A, Cicciù M, Matacena G, Cervino G, Troiano G, Zhurakivska K, Laino L. Postextraction Dental Implant in the Aesthetic Zone, Socket Shield Technique Versus Conventional Protocol. J Craniofac Surg. 2018, 29, 1037-1041. Habashneh RA, Walid MA, Abualteen T, Abukar M. Socket-shield Technique and Immediate Implant Placement for Ridge Preservation: Case Report Series with 1-year Follow-up. J Contemp Dent Pract. 2019, 20, 1108-1117.). On the contrary, Siormpas et al. suggest that it is not necessary to graft the space between the residual vestibular root portion and the implant (Siormpas KD, Mitsias ME, Kontsiotou-Siormpa E, Garber D, Kotsakis GA. Immediate implant placement in the esthetic zone utilizing the "root-membrane" technique: clinical results up to 5 years postloading. Int J Oral Maxillofac Implants. 2014, 29, 1397-405.). This concept is supported by recent histological data showing that without the use of biomaterials, new bone grows in the space between the dentin fragment and the implant (Bramanti E, Norcia A, Cicciù M, Matacena G, Cervino G, Troiano G, Zhurakivska K, Laino L. Postextraction Dental Implant in the Aesthetic Zone, Socket Shield Technique Versus Conventional Protocol. J Craniofac Surg. 2018, 29, 1037-1041.).
Reviewer 2: What are the results of other systematic reviews on the subject?
Response: In order to adapt to the reviewer's 1 comments, we clarify that there are one systematic review published in 2017: Gharpure AS, Bhatavadekar NB. Current Evidence on the Socket-Shield Technique: A Systematic Review. J Oral Implantol. 2017 Oct;43(5):395-403; which included: 1 clinical case-control study, 4 animal histological reports, 1 clinical abstract, and 17+2* case reports. Other in 2019: Mourya A, Mishra SK, Gaddale R, Chowdhary R. Socket-shield technique for implant placement to stabilize the facial gingival and osseous architecture: A systematic review. J Investig Clin Dent. 2019 Nov;10(4):e12449; which included 11 case reports, 6 case series, 1 human randomized control trial (RCT), 1 technical report and 2 animal RCT. Other in 2020: Blaschke C, Schwass DR. The socket-shield technique: a critical literature review. Int J Implant Dent. 2020 Sep 7;6(1):52. And two in 2021: Sáez-Alcaide LM, Fernández-Tresguerres FG, Brinkmann JC, Segura-Mori L, Iglesias-Velázquez O, Pérez-González F, López-Pintor RM, García-Denche JT. Socket shield technique: A systematic review of human studies. Ann Anat. 2021 Jun 1:151779 and Ogawa T, Sitalaksmi RM, Miyashita M, Maekawa K, Ryu M, Kimura-Ono A, Suganuma T, Kikutani T, Fujisawa M, Tamaki K, Kuboki T. Effectiveness of the Socket Shield Technique in Dental Implant: A Systematic Review. J Prosthodont Res. 2021 Mar 9. doi: 10.2186/jpr.JPR_D_20_00054. Epub ahead of print; which included articles with a follow-up between 3-60 months after placement. However, none of the systematic reviews include a meta-analysis (quantitative analysis). The main results (qualitative) reported in this articles were that socket-shield technique achieved a good esthetic appearance, the failure rate was low without complications, although there were some failures due to failed implant osseointegration, socket shield mobility and infection, socket shield exposure, socket shield migration, and apical root resorption.
Reviewer 2: Figures: Please improve quality and resolution.
Response: In order to adapt to the reviewer's 2 comments, we have improved the quality and resolution of the figures.
Reviewer 2: Abbreviations: Insert a summary of abbreviations used in the text prior to “Reference” section.
Response: In order to adapt to the reviewer's 2 comments, we have added an “Abbreviations” section.
We take this opportunity to thank the recommendations and suggestions made by the reviewer to improve the document.
Yours sincerely,
Reviewer 3 Report
I appreciate the opportunity to review the article entitled “Failure Rate, Marginal Bone Loss and Pink Esthetic of Socket-Shield Technique for Immediate Dental Implant Placement in the Esthetic Zone. A Systematic Review and Meta-Analysis”. Socket-shield technique is a relatively new operation procedure for immediate implant placement. The technique sounds good in theory, but the established evaluation has yet to be reached. So, this manuscript is interesting for implant surgeons. I put some points I have noticed below. Hope this helps.
1. Please indicate the causes of failure available from the literature. Was socket-shield technique directly related to the failures?
2. It’s easy to guess that the marginal bone loss could affect PES(pink esthetic score). How was the marginal bone loss statistically related to PES in your study?
3. The following systematic reviews on socket-shield technique have recently been published. Please discuss your data comparing to these reviews.
1)Blaschke, C., Schwass, D.R. The socket-shield technique: a critical literature review. Int J Implant Dent 6, 52 (2020). https://doi.org/10.1186/s40729-020-00246-2
2)Ogawa T, Sitalaksmi RM, et al. Effectiveness of the Socket Shield Technique in Dental Implant: A Systematic Review. J Prosthodont Res. 2021 Mar 9. doi:10.2186/jpr.JPR_D_20_00054. Epub ahead of print. PMID: 33692284.
Author Response
Dear Ms Fancy Yan:
I’m pleased to resubmit the manuscript of the work entitled, “Failure Rate, Marginal Bone Loss and Pink Esthetic of Socket-Shield Technique for Immediate Dental Implant Placement in the Esthetic Zone. A Systematic Review and Meta-Analysis“
Reviewer 3: Moderate English changes required
Response: In order to adapt to the reviewer's 2 comments, we have send the manuscript to the English Editing Service of MDPI. We attached the Certificate.
Reviewer 3: Please indicate the causes of failure available from the literature. Was socket-shield technique directly related to the failures?
Response: In order to adapt to the reviewer's 3 comments, we clarify that the lack of osteintegration, infection, mobilization, migration and resorption of the root fragment, have been highlighted as possible causes of failure of the socket-shield technique. Baumer et al described a case of apical resorption of the fragment, possibly due to the presence of microbiological remains at the root apex (Bäumer D, Zuhr O, Rebele S, Hürzeler M. Socket Shield Technique for immediate implant placement - clinical, radiographic and volumetric data after 5 years. Clin Oral Implants Res. 2017 Nov;28(11):1450-1458). Habashneh et al described the formation of an 8mm pocket in one root and the exposure of the other (Habashneh RA, Walid MA, Abualteen T, Abukar M. Socket-shield Technique and Immediate Implant Placement for Ridge Preservation: Case Report Series with 1-year Follow-up. J Contemp Dent Pract. 2019 Sep 1;20(9):1108-1117.). Gluckman et al reported lack of osteintegration in 5 implants, 5 infection and mobilization of the fragment, 13 internal exposure (towards the restoration) possibly due to lack of adequate space between the coronal portion of the segment and the subgingival contour of the crown; 4 external exposure (towards the oral cavity) and 1 presented fragment migration (Gluckman H, Du Toit J, Salama M. The Pontic-Shield: Partial Extraction Therapy for Ridge Preservation and Pontic Site Development. Int J Periodontics Restorative Dent. 2016 May-Jun;36(3):417-23.).
Reviewer 3: It’s easy to guess that the marginal bone loss could affect PES(pink esthetic score). How was the marginal bone loss statistically related to PES in your study?
Response: In order to adapt to the reviewer's 3 comments, we clarify that the marginal bone loss and PES were analyzed independently, because there are more factors that could influence the PES and it could bias the study.
Reviewer 3: The following systematic reviews on socket-shield technique have recently been published. Please discuss your data comparing to these reviews: Blaschke, C., Schwass, D.R. The socket-shield technique: a critical literature review. Int J Implant Dent 6, 52 (2020). https://doi.org/10.1186/s40729-020-00246-2. Ogawa T, Sitalaksmi RM, et al. Effectiveness of the Socket Shield Technique in Dental Implant: A Systematic Review. J Prosthodont Res. 2021 Mar 9. doi:10.2186/jpr.JPR_D_20_00054. Epub ahead of print. PMID: 33692284
Response: In order to adapt to the reviewer's 3 comments, we have discussed the data with our findings.
We take this opportunity to thank the recommendations and suggestions made by the reviewer to improve the document.
Yours sincerely,